# Transitioning Representations between Languages for Cross-lingual Event Detection via Langevin Dynamics

**Chien Van Nguyen**[1], **Huy Huu Nguyen**[2],
**Franck Dernoncourt**[3], and **Thien Huu Nguyen**[1,2]
[1]Department of Computer Science, University of Oregon, Eugene, OR, USA
[2]VinAI Research, Vietnam
[3] Adobe Research, USA
{chienn,thien@cs}@uoregon.edu
v.huynh16@vinai.io, franck.dernoncourt@adobe.com

## Abstract

Cross-lingual transfer learning (CLTL) for event detection (ED) aims to develop models in high-resource source languages that can be directly applied to produce effective performance for lower-resource target languages. Previous research in this area has focused on representation matching methods to develop a language-universal representation space into which source- and target-language example representations can be mapped to achieve cross-lingual transfer. However, as this approach modifies the representations for the source-language examples, the models might lose discriminative features for ED that are learned over training data of the source language to prevent effective predictions. To this end, our work introduces a novel approach for cross-lingual ED where we only aim to transition the representations for the target-language examples into the source-language space, thus preserving the representations in the source language and their discriminative information. Our method introduces Langevin Dynamics to perform representation transition and a semantic preservation framework to retain event type features during the transition process. Extensive experiments over three languages demonstrate the state-of-the-art performance for ED in CLTL.

## 1 Introduction

Event Detection (ED) is one of the fundamental problems in Information Extraction (IE). This task aims to identify and classify the words or phrases that most clearly evoke events in text (called event triggers). For instance, given a sentence "*He was* **fired** *from the corporation yesterday.*", an ED system needs to recognize the word "*fired*" as an event trigger of the type *Attack*. With the advances in deep learning, recent research has produced impressive performance for ED (Nguyen and Grishman, 2015; Liu et al., 2017; Yang et al., 2019; Liu et al., 2020; Tran Phu and Nguyen, 2021).

However, despite such progress, existing datasets for ED are limited to a small set of popular languages (e.g., English and Chinese) due to the high cost of data annotation for this task. The lack of annotated data has thus hindered the development of ED systems for a larger set of languages and the benefits of the technology for broader society. To enable ED for more languages while avoiding expensive annotation efforts, recent research has focused on zero-shot cross-lingual transfer learning (CLTL) for ED where models are trained on annotated data of a high-resource *source* language and directly applied to extract events for text in another low-resource *target* language (Nguyen et al., 2021; Guzman Nateras et al., 2023; Nguyen et al., 2023). In this setting, labeled data is not available for the target language; however, unlabeled data can be used to facilitate knowledge transfer between languages.

The key challenging in CLTL for ED is to bridge the gap between representation spaces for the source and target languages to enable knowledge transfer. As such, previous work in this area has focused on the representation matching approaches, aiming to transform the representations for the source and target languages into a common space to train language-general ED models over source language data, e.g., via similarity regularization (Nguyen et al., 2021) or adversarial learning (Guzman-Nateras et al., 2022). However, to achieve a common space, the source-language representations might need to sacrifice language-specific information, which can involve discriminative features for event prediction due to the mix of information in the representations for input text. Consequently, losing discriminative features might lead to impaired performance for ED.

To this end, our work explores a new direction for CLTL for ED using representation transition. Instead of transforming the source-language representations, which might eliminate discriminative

features for ED, we only seek to adapt the target-language representations into the source space whose results can be directly fed into the source-language models for event prediction. In this way, we can preserve the source-language representations learned over training data for ED to maximize prediction ability. To achieve this idea, we propose to employ an energy function to model the representations for the source-language training examples where low-energy regions correspond to high likelihood of belonging to the source language (LeCun et al., 2006). Afterward, target-language representations can be adapted to the source-language space by transitioning them into low-energy regions (i.e., minimizing the energy function). As such, our method leverages Langevin Dynamics (Welling and Teh, 2011; Du and Mordatch, 2019) to perform efficient representation transition for the target language for cross-lingual ED where gradient descent is utilized to iteratively update the initial target representations into the source language with low-energy regions. To our knowledge, this is the first work to explore energy functions and Langevin Dynamics for CLTL in IE and ED.

Finally, we introduce a mechanism to regulate the adapted representations for the target language examples from Langevin Dynamics so they can maintain semantic similarity of event types with the original representations. This regularization is necessary to ensure that the predictions of the source language model over the adapted representations are correct for the target examples. Extensive evaluations over three languages demonstrate our state-of-the-art performance for cross-lingual ED.

## 2 Model

**Base Model**: Following prior work (Guzman-Nateras et al., 2022), we formulate ED as a sequence labeling problem to facilitate cross-lingual transfer learning (CLTL). Given an input sentence of $n$ tokens $W = \{w_1, w_2, \ldots, w_n\}$, our ED model needs to predict its corresponding sequence of labels $Y = \{y_1, y_2, \ldots, y_n\}$. Here, $y_i$ is the label for the token $w_i$ using the BIO annotation schema to capture event mentions and types in $W$.

In our model, we first use the multilingual pre-trained language model XLM-R (Conneau et al., 2019) as the feature extractor for the input sentence $W$, leading to a representation vector $h_i$ for each word $w_i$: $h_1, h_2, \ldots, h_n = $ XLM-R$_{\theta_X}(w_1, w_2, \ldots, w_n)$. Here, $\theta_X$ captures

the parameters of XLM-R. Afterward, the representation $h_i$ is fed into a feed-forward network (FFN) with parameters $\theta_C$ to produce a label distribution for $w_i$ for event prediction. For convenience, in the following, we will use the subscripts $src$ and $tgt$ to specify words and representations from the source and target languages (respectively).

**Energy Model and Langevin Dynamics**: As discussed above, to perform CLTL for ED, we propose to learn an energy function (LeCun et al., 2006) to model the representation distribution of the source-language examples. The representations of the target-language examples can then be transitioned toward this source distribution to perform ED. Formally, in our method, an energy function $E_{\theta_E}(h) : \mathbb{R}^d \to \mathbb{R}$ is implemented as a feed-forward network with parameters $\theta_E$ to map a representation $h$ to a scalar. This function can then be used to define a distribution over the representation space: $p_{\theta_E}(h) = \frac{\exp(-E_{\theta_E}(h))}{\mathcal{Z}_{\theta_E}}$, where $\mathcal{Z}_{\theta_E} = \int \exp(-E_{\theta_E}(h))dh$ denotes the normalizing term. As such, to draw a sample from this distribution, we can employ the Langevin Dynamics process (Du and Mordatch, 2019): $h^{k+1} \leftarrow h^k - \frac{\epsilon}{2}\nabla_{h^k}E(h^k) + \omega$, where $\omega \sim \mathcal{N}(0, \sigma)$ is a Gaussian noise and $\epsilon$ denotes the step size. Here, the initial representation $h^0$ will be sampled from an uniform distribution, and then transitioned along the direction of the gradient of $E_{\theta_E}$ as in gradient descent. After $K$ steps, the resulting representation $h^K$ is expected to have low energy, thus having higher chance to be distributed from $p_{\theta_E}(h)$.

**Training**: Given the training data in the source language, our method trains an ED model and an energy function for the source examples in two steps. In the first step, the base ED model is trained using the negative log-likelihood loss, optimizing the parameters $\theta_X$ and $\theta_C$ for XLM-R and FFN: $\mathcal{L}_{base}(\theta_X, \theta_C) = -\sum_{i=1}^{n} \log(y_i^{src}|w_i^{src}, \theta_F, \theta_C)$.

In the second step, we freeze the parameters $\theta_X$ for XLM-R and train the energy function $E_{\theta_E}$ by optimizing the negative log-likelihood for the induced representations $h_i$ for $w_i \in W$ in the source language: $\mathcal{L}_{eny} = -\sum_{j\in L} \log P_{\theta_E}(h_j^{src})$, where $L$ involves indexes for the words in the event trigger spans in $W$. Here, although computing the normalizing term $\mathcal{Z}_{\theta_E}$ for $E$ is intractable, the gradient of $\log P_{\theta_E}(h)$ can still be estimated to perform training in our second step (Du and Mordatch, 2019; Song and Kingma, 2021): $\nabla_{\theta_E} \log P_{\theta_E}(h) = \mathbb{E}_{P_{data}(h)}[-\nabla_{\theta_E}E_{\theta_E}(h)] + \mathbb{E}_{P_{\theta_E}(h)}[\nabla_{\theta_E}E_{\theta_E}(h)]$.

As such, the data for the first term is drawn from the representations for the source language examples $P_{data}(h)$ while those for the second term can be sampled from the model representation distribution $P_{\theta_E}(h)$ using our Langevin Dynamics process.

At inference time for CLTL, given an input sentence $W^{tgt}$ in the target language, the representation $h_i^{tgt}$ for each word $w_i^{tgt} \in W^{tgt}$ from XLM-R is iteratively adapted to the source space using $K$-step Langevin Dynamics, resulting in the adapted representation $\overline{h}_i^{tgt}$. Afterward, due to their compatibility, we can apply the source-trained ED model $\theta_C$ over $\overline{h}_i^{tgt}$ to predict the event type for $w_i^{tgt}$.

**Semantic Preservation**: Although Langevin Dynamics can adapt the representation $h_i^{tgt}$ to achieve features in the source language, it cannot guarantee that the adapted representation $\overline{h}_i^{tgt}$ can preserve semantic information of event types from $h_i^{tgt}$ to produce correct prediction based on $\overline{h}_i^{tgt}$. To facilitate semantic maintenance in the adaption, we propose to decompose a representation $h$ into language-invariant and language-specific components $z$ and $l$ (respectively). As such, our intuition for semantic maintenance is that the target-to-source adaptation process should retain language-invariant information while eliminating target-specific features and absorbing source-language features along the way. We employ both labeled data in the source language and unlabeled data in the target language to achieve semantic preservation in this step.

**Representation Decomposition**: To disentangle the representations, we employ the Variationl Auto-Encoder (VAE) framework (Kingma and Welling, 2014) to introduce two encoders $q_{\phi_z}(z|h)$ and $q_{\phi_l}(l|h)$ to transform a representation $h$ into stochastic dimensional spaces for language-invariant and language-specific representations $z$ and $l$ respectively. In addition, a decoder $p_{\phi_h}(h|z,l)$ is introduced to infer the representation $h$ from $z$ and $l$. Here, the distributions $q_{\phi_z}(z|h)$, $q_{\phi_l}(l|h)$, and $p_{\phi_h}(h|z,l)$ are assumed to be Gaussian and feed-forward neural networks with parameters $\phi_z$, $\phi_l$, and $\phi_h$ are leveraged to compute the means and variances for the distributions from corresponding representations. To this end, VAEs learn the parameters by minimizing the negation of the variational lower bound:
$$\mathcal{L}_{VAE} = -\mathbb{E}_{q_{\phi_z}(z|h)q_{\phi_l}(l|h)}\left[\log p_{\phi_h}(h|z,l)\right] + KL(q_{\phi_z}(z|h)||p(z)) + KL(q_{\phi_l}(l|h)||p(l)),$$ where the first term is the reconstruction loss, $p(z)$ and $p(l)$ are the standard Gaussian distribution, and

the Kullback-Leibler (KL) divergences serve as the regularization. This loss can be obtained from unlabeled data for both source and target languages.

For the language-invariant representation, we expect $z$ to encode information on event types to allow transferability across languages. As such, we utilize $z$ to predict the event type $y$ using training examples in the source language. In particular, a feed-forward network is introduced to compute the distribution $P(y|z)$ from $z$ and the log-likelihood $\mathbb{E}_{q_{\phi_z}(z|h)}[\log P(y|z)]$ is used to train the network. For the representation $l$, its language-specific information is achieved by computing the distribution $P(d|l)$ with a feed-forward network $FF^L$ to predict the language identity $d$ for $h$ (i.e., $d \in \{source, target\}$). $FF^L$ can then be trained with the objective $\mathbb{E}_{q_{\phi_l}(l|h)}[\log P(d|l)]$ using unlabeled data in both source and target languages. Consequently, to promote the expected features for $z$ and $l$, we minimize the combined function: $\mathcal{L}_{dec} = -\mathbb{E}_{q_{\phi_z}(z|h)}[\log P(y|z)] - \mathbb{E}_{q_{\phi_l}(l|h)}[\log P(d|l)]$.

| Model | Langauge Pairs | | | | | |
|---|---|---|---|---|---|---|
| Source | EN | EN | ZH | ZH | AR | AR |
| Target | ZH | AR | EN | AR | EN | ZH |
| BERT-CRF | 68.5 | 30.9 | 37.5 | 20.1 | 40.1 | 58.8 |
| BERT-CRF-LAT | 70.0 | 33.5 | 41.2 | 20.3 | 37.2 | 55.6 |
| BERT-CRF-FTUT | 69.4 | 33.4 | 42.9 | 20.0 | 36.5 | 56.3 |
| BERT-CRF-CCCAR | 72.1 | 42.7 | 45.8 | 20.7 | 40.7 | 59.8 |
| XLMR-CRF | 70.5 | 43.5 | 41.7 | 32.8 | 45.4 | 61.8 |
| XLMR-CRF-LAT | 70.2 | 43.4 | 42.3 | 33.2 | 45.2 | 60.9 |
| XLMR-CRF-FTUT | 71.1 | 43.7 | 42.1 | 32.9 | 45.9 | 62.1 |
| XLMR-CRF-CCCAR | 74.4 | 44.1 | 49.5 | 34.3 | 46.3 | 62.9 |
| XLMR-OACLED | 74.6 | 44.9 | 45.8 | 35.1 | 48.0 | 63.1 |
| **RepTran (ours)** | **77.7** | **46.6** | **50.6** | **39.5** | **50.8** | **66.2** |
| -$\mathcal{L}_{dec}$ | 74.1 | 43.7 | 46.2 | 36.6 | 45.4 | 63.8 |
| -$\mathcal{L}_z$ | 76.5 | 45.1 | 48.4 | 37.2 | 48.7 | 64.3 |
| -$\mathcal{L}_l$ | 75.2 | 43.9 | 47.3 | 36.1 | 46.5 | 63.4 |
| - sem. preservation | 74.3 | 42.9 | 46.6 | 35.7 | 45.2 | 62.9 |

Table 1: Cross-lingual performance (F1 scores) on test data. Each column corresponds to one language pair where source languages are shown above the target languages. The proposed model is significantly better than other models with $p < 0.01$.

**Representation Constraints**: Given the representation $h^{tgt}$ for a word in the target language, we expect that its language-invariant features for event types in $z^{tgt}$ will be preserved in the adapted representation $\overline{h}^{tgt}$ from Langevin Dynamics to enable successful ED with the source language model. To implement this idea, we first compute the language-invariant components $z^{tgt}$ and $\overline{z}^{tgt}$ from $h^{tgt}$ and its adapted version $\overline{h}^{tgt}$ via: $z^{tgt} \sim q_{\phi_z}(z|h^{tgt})$ and $\overline{z}^{tgt} \sim q_{\phi_z}(z|\overline{h}^{tgt})$. Afterward, the maintenance of language-invariant information is enforced by minimizing the $L_2$ difference between $z^{tgt}$ and $\overline{z}^{tgt}$, leading to the objective: $\mathcal{L}_z = ||z^{tgt} - \overline{z}^{tgt}||_2^2$.

In addition, to realize the target-to-source transition of $h^{tgt}$ to $\overline{h}^{tgt}$, we expect that the language-

specific component of $\bar{h}^{tgt}$, i.e., $\bar{l}^{tgt} \sim q_{\phi_l}(l|\bar{h}^{tgt})$, should be predictive for the source language. To this end, we update our model so the language prediction network $FF^L$ can predict the source language from $\bar{l}^{tgt}$. As such, we obtain the distribution $P(d|\bar{l}^{tgt})$ from $FF^L$ and minimize the likelihood loss: $\mathcal{L}_l = -\mathbb{E}_{q_{\phi_l}(l|\bar{h}^{tgt})}[P(source|\bar{l}^{tgt})]$.

Finally, the overall loss to train our model in the second step is: $\mathcal{L} = \mathcal{L}_{eny} + \mathcal{L}_{VAE} + \mathcal{L}_{dec} + \mathcal{L}_z + \mathcal{L}_l$.

## 3 Experiments

**Datasets and Hyper-parameters**: Following previous work on CLTL for ED (M'hamdi et al., 2019; Nguyen et al., 2021), we evaluate our proposed method (called RepTran) on the dataset ACE05 (Walker et al., 2006). This dataset contains documents in three languages: English (EN), Chinese (ZH) and Arabic (AR) with annotations for 33 event types. The same data split and preprocessing as in previous work (Nguyen et al., 2021) are employed for a fair comparison. For each language, the data split provides training, development, and test portions for experiments. Given the three languages, we consider six possible pairs of languages to form the source and target languages in our cross-lingual transfer learning experiments. For each language pair, we train the models on the training data of the source language and evaluate the performance on the test data of the target language. Unlabeled data for the target language is obtained by removing labels from its training data portion as done in previous work (Nguyen et al., 2021).

We utilize the XLM-R base model with 768 dimensions in the hidden vectors to be comparable with previous work (Nguyen et al., 2021; Guzman-Nateras et al., 2022). We tune the hyper-parameters for our model over the development data using the EN→ZH language pair. The same hyper-parameters are employed to train models for other language pairs in our experiments. In particular, the following values are selected for our model: 2 layers for the feed-forward networks with 768 dimensions for the hidden layers, $3e$-5 for the AdamW optimizer, 16 for the batch size, and $K = 128$ for the number of update steps, $\epsilon = 5$ for the step size in the update rule, and $\sigma = 0.005$ for the variance in the Gaussian noise for the target-language representation adaptation with Langevin Dynamics.

**Baselines**: We compare our method with a variety of baselines for CLTL for ED, based on multi-lingual language models, i.e., mBERT-CRF and XLMR-CRF (Nguyen et al., 2021), fine-tuning with unlabeled target data (FTUT), i.e., BERT-CRF-FTUT and XLMR-CRF-FTUT (Nguyen et al., 2021), representation matching, i.e., BERT-CRF-CCCAR and XLMR-CRF-CCCAR (Nguyen et al., 2021), and adversarial training, i.e., BERT-CRF-LAT, XLMR-CRF-LAT (Nguyen et al., 2021), and XLMR-OACLED (Guzman-Nateras et al., 2022). As such, XLMR-CRF-CCCAR and XLMR-OACLED represent the current state-of-the-art methods for CLTL for ED over ACE05.

**Evaluation**: Table 1 presents the cross-lingual performance of the models. The most important observation from the table is that the proposed method RepTran is significantly better than the baselines across all six language pairs, thus clearly demonstrating the benefits of the Langevin-based representation transition for cross-lingual ED. In addition, to evaluate the contribution of the proposed semantic preservation, we perform an ablation study where each introduced loss (i.e., $\mathcal{L}_{dec}$, $\mathcal{L}_z$, and $\mathcal{L}_l$) is excluded from the training process. We also explore the complete removal of the semantic preservation component. It is clear from the table that removing any of the components would hurt the performance of RepTran significantly, thus confirming the importance of the semantic preservation techniques in our model.

**Representation Visualization**: To illustrate the Lengevin-based representation adaptation process from the target to the source language space in Rep-Tran, we visualize the representations of the event triggers in the target language along its adaptation process. In particular, using English and Chinese as the source and target languages, Figure 1 shows the t-SNE representation visualization for a sample of target-language event triggers for the most frequent event types at different steps in the adaptation process. It is clear from the figure that the target language representations are gradually shifted toward the source language space based on Langevin Dynamics and the energy function (i.e., the triangles become closer and closer to the circles). Importantly, due to the introduction of our semantic preservation mechanism, the final states of the target language representations (i.e., $K = 128$) can also preserve their information about event types as demonstrated by the proximity of the circles and triangles of the same colors for event types.

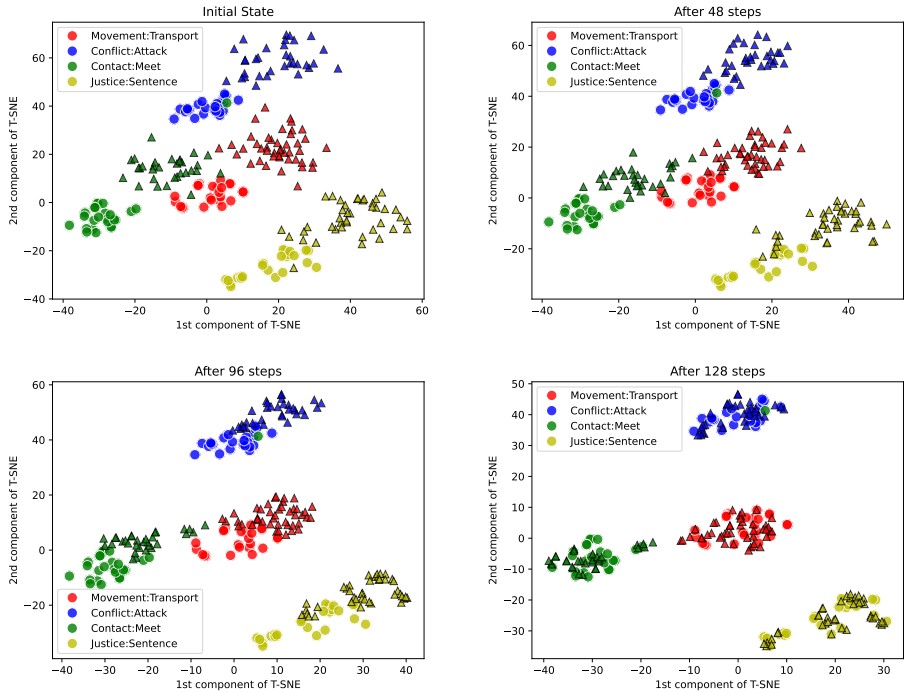

Figure 1: T-SNE visualization for the representation transition process from Chinese event triggers (i.e., the target) to the English language space (i.e., the source) based on Langevin Dynamics and the energy function. Circles and triangles indicate representations for the English and Chinese examples respectively while colors encode event types.

## 4 Related Work

A large body of previous ED research is dedicated to monolingual learning, i.e., training and testing over the same languages (Nguyen et al., 2016; Yang and Mitchell, 2016; Lu and Nguyen, 2018; Lai et al., 2020; Lin et al., 2020). The models might consider different domains (Nguyen and Nguyen, 2018; Man Duc Trong et al., 2020). To address the language barriers, recent work has studied CLTL for ED, focusing on mapping representations for different languages into the same space for transfer learning, e.g., using bilingual resources (Muis et al., 2018; Liu et al., 2019), multilingual language models (M'hamdi et al., 2019), and adversarial training (Guzman-Nateras et al., 2022). Our work is also related to previous work on data projection for ED and IE where texts in the target language are translated and aligned into the source language for prediction at test time (Riloff et al., 2002; Yarmo-hammadi et al., 2021; Yu et al., 2023). However, data projection requires high-quality machine translation and alignment systems, which might not be available for different languages. Our method is different as we project representations (not actual texts) in the target to the source language space to avoid translation systems.

## 5 Conclusion

We introduce a novel method for CLTL for ED where representations of examples in the target language are transferred into the source language space at test time for prediction with the source language model. Our extensive experiments confirm the benefits of the proposed method. In the future, we plan to extend our method to other tasks in IE.

## Acknowledgement

This research has been supported by the Army Research Office (ARO) grant W911NF-21-1-0112, the NSF grant CNS-1747798 to the IUCRC Center for Big Learning, and the NSF grant # 2239570. This research is also supported in part by the Office of the Director of National Intelligence (ODNI), Intelligence Advanced Research Projects Activity (IARPA), via the HIATUS Program contract 2022-22072200003. The views and conclusions contained herein are those of the authors and should not be interpreted as necessarily representing the official policies, either expressed or implied, of ODNI, IARPA, or the U.S. Government. The U.S. Government is authorized to reproduce and distribute reprints for governmental purposes notwithstanding any copyright annotation therein.

## Limitations

In this work, we develop a novel cross-lingual transfer learning method for event detection that achieves state-of-the-art performance on benchmark datasets for this problem. However, our work still has several limitations that can be further explored in future work. First, our RepTran method leverages Langevin Dynamics to perform representation transition for target language examples at inference time to achieve knowledge transfer. While our experiments demonstrate the superior performance of this approach for cross-lingual ED, our method does involve an additional computational cost at inference time for representation transition. In Appendix A, we show that this additional cost is not substantial and it can be acceptable given the demonstrated benefits. Nonetheless, we believe that future work can explore this direction further to reduce computational costs at inference time for RepTran to improve its efficiency and applications. Second, our method relies on the multilingual pre-trained language model XLMR to encode text in different languages. Although XLMR can support more than 100 languages, the world has more than 7000 languages and our method cannot be applied directly to many other languages. Extending the language set of pre-trained language models is an important direction that can improve our method. Finally, to perform semantic preservation, our methods need to employ unlabeled data in the target language. While unlabeled data can be more available for many languages, it might still be challenging to retrieve text for extremely low-resource languages. As such, future research can explore methods to relax this requirement of unlabeled data for the target language to maximize the applicability of cross-lingual transfer learning methods for ED.

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

## A Speed Inference Evaluation

Our Langevein-based representation transition method needs to adapt each target language example at inference time into the source language space to allow event prediction with the source language ED model. As such, compared to previous cross-lingual ED method, in addition to the encoding cost for a target language example at inference time, our method needs to perform an additional step to transition the representations into the source language space based on the energy function. This implies an additional computational cost for predicting each target language example.

| Model | Inference Time |
|---|---|
| XLMR-CRF | 1.00x |
| RepTran ($K = 64$) | 1.15x |
| RepTran ($K = 128$) | 1.24x |

Table 2: Latency cost for our RepTran model. All results are computed with a single NVIDIA V100 GPU.

To measure the impact of this additional cost for the inference time of our method, Table 2 compares the inference time of the baseline XLMR-CRF and our model RepTran over test data. Using the inference time of XLMR-CRF as the reference, we show the inference time for adaptation steps at both $K = 64$ and $K = 128$ (the final step). Although the representation transition does increase the inference time for RepTran, the additional cost is not substantial and can be acceptable to achieve greater cross-lingual performance for ED.