# OpenReview forum: "Transitioning Representations between Languages for Cross-lingual Event Detection via Langevin Dynamics"
_EMNLP/2023/Conference — EMNLP 2023 Findings_

### Official Review · Reviewer_87k9 · 2023-08-04

**Soundness:** 3

**Excitement:**

3: Ambivalent: It has merits (e.g., it reports state-of-the-art results, the idea is nice), but there are key weaknesses (e.g., it describes incremental work), and it can significantly benefit from another round of revision. However, I won't object to accepting it if my co-reviewers champion it.

**Paper Topic And Main Contributions:**

This paper presents a model for cross-lingual event detection, which relies on pre-trained multilingual language model to generate representations for source and target languages, and then utilizes Langevin Dynamics to transition the representations of the target language to source language. The paper also proposes several components to restrict the transitions.

The main contribution in the paper is the proposal of using the Langevin Dynamics to transition the representations of the target language to source language. Combined with several other methods, the proposed model outperforms baselines on ACE05 datasets.

**Reasons To Accept:**

1. The conducted experiments demonstrate the effectiveness of the proposed model.
2. The ideas can be adapted to other cross-lingual tasks.

**Reasons To Reject:**

1. The motivation is not very convincing. The paper claims that it aims to transition the representations for the target-language examples into the source-language space. However the proposed model relies on pre-trained multilingual language model to generate the representations for target language and source language. The representations of different languages generated by XLM-R are already projected into a common space. Langevin Dynamics plays a role of matching the representation distributions of the target language data and the source language data. Thus some Generative Adversarial Networks related work should be mentioned and included as baseline.
2. The Semantic Preservation components proposed in the paper can also be incorporated into other models. The proposed model does not significantly outperforms baselines with the semantic preservation components removed.

**Reproducibility:**

3: Could reproduce the results with some difficulty. The settings of parameters are underspecified or subjectively determined; the training/evaluation data are not widely available.

**Reviewer Confidence:**

4: Quite sure. I tried to check the important points carefully. It's unlikely, though conceivable, that I missed something that should affect my ratings.

---

> ### Author Rebuttal · Authors · 2023-08-29
>
> Thank you for your comments and suggestions. Please find below our responses for your questions and concerns.
>
> **Reviewer**: *"... the proposed model relies on pre-trained multilingual language model to generate the representations for target language and source language. The representations of different languages generated by XLM-R are already projected into a common space. Langevin Dynamics plays a role of matching the representation distributions of the target language data and the source language data. Thus some Generative Adversarial Networks related work should be mentioned and included as baseline."*
>
> **Our Response**:
>
> Thank you for your comment. However, we would like to note that although XLM-R could project the representations of the source and target languages into a common space, there are still significant gaps between source- and target-language representations as demonstrated in Figure 1 of our Appendix B. Such representation gaps can cause significant challenges for cross-lingual knowledge transfer, necessitating additional alignment steps to bridge the gaps between the source and target representations to boost the performance. This performance gap issue from XLM-R representations for source and target languages have also been observed in previous cross-lingual work in this area, e.g., [1].
>
> To achieve the representation alignment for the source and target languages, we propose a novel representation transition approach that aims to transition the representations from the target languages into the source language space, thus preserving the source-language representations and their discriminative features to improve event detection. As such, we employ Langevin Dynamics and an energy function whose representation sampling can be used to implement the target-to-source representation transition process. We agree with the reviewer that Generative Adversarial Networks (GANs) are relevant to our approach as they also aim to match the representation distributions. However, we would like to highlight a crucial distinction between our method and adversarial-based approaches. Our method is designed to preserve the source-language representations and their language-specific and discriminative features learned over training data, which can be valuable for the event detection task. In contrast, adversarial training, when applied to cross-lingual transfer learning for event detection, will perform elimination of language-specific features in both the source and target language representations to achieve a shared space for transfer leraning. As discussed in our introduction (L69-76), removing language-specific features might also lead to the exclusion of discriminative features for event detection, thus hindering the performance of the methods.
>
> Furthermore, we would like to clarify that we have already provided a comprehensive comparison of our model with a range of adversarial-based methods, such as the *-LAT, *-CCCAR, and *-OACLED models in Table 1. These results demonstrate strong performance for our representation transition method over adversarial learning models, thus confirming the effectiveness of our model for cross-lingual event detection.
>
> [1] Nguyen et al., 2021. Crosslingual transfer learning for relation and event extraction via word category and class alignments. In EMNLP 2021.
>
> **Reviewer**: *"The Semantic Preservation components proposed in the paper can also be incorporated into other models. The proposed model does not significantly outperforms baselines with the semantic preservation components removed."*
>
> **Our Response**:
>
> Thank you for your comment. We agree with the reviewer that the Semantic Preservation component is critical for the performance of our model as demonstrated in our ablation study. However, we would like to emphasize that this Semantic Preservation component is inherent to our model as it is specially designed for the representation transition approach, which involves a sequence of representations sampled with Langevin Dynamics to transition a representation vector in the target space to the source space. As such, our representation transition method involves an initial representation computed over an example in the target language with XLMR and a final state representation from Langevin Dynamics to serve as the adapted representation to the source space. The representation contraint is then introduced to ensure that the event type information in these initial and adapted representations of the target examples are preserved in our model. To this end, we believe the Semantic Preservation component cannot be applied directly to previous cross-lingual methods for event detection as they don't involve any the representation transition process with initial and adapted representations, which are necessary to perform event type information preservation. We hope the reviewer can reconsider this point for us.

---

### Official Review · Reviewer_9TYZ · 2023-08-05

**Soundness:** 3

**Excitement:**

3: Ambivalent: It has merits (e.g., it reports state-of-the-art results, the idea is nice), but there are key weaknesses (e.g., it describes incremental work), and it can significantly benefit from another round of revision. However, I won't object to accepting it if my co-reviewers champion it.

**Paper Topic And Main Contributions:**

The paper, "Transitioning Representations between Languages for Cross-lingual Event Detection via Langevin Dynamics", introduces a new method, RepTran, for cross-lingual event detection. The authors claim that their method outperforms existing ones on the ACE05 dataset. However, the paper lacks a comprehensive analysis of the methods used and the experimental results, raising questions about the validity and applicability of the proposed method.

**Questions For The Authors:**

1. How does the proposed method compare with existing methods in terms of computational efficiency, especially considering the additional computational cost at inference time for representation transition?
2. Could you elaborate on the limitations of the results? Were there any surprising findings or anomalies in the results that were not discussed in the paper?
3. Could you provide more insights into why the proposed method outperforms existing methods? What specific features or aspects of the method contribute to its superior performance?
4. Given the complexity of the proposed method, how do you see its practical application, especially in real-world scenarios where computational resources might be limited?
7. Are there any specific future research directions you suggest based on the findings of this paper?

**Reasons To Accept:**

1. The paper attempts to address a significant problem in the field of cross-lingual event detection.
2. The authors provide a novel approach to the problem, which could potentially contribute to the field.

**Reasons To Reject:**

1. The authors do not provide a thorough comparison with existing methods. This leaves the reader unsure of how the proposed method stands against current state-of-the-art methods.
2. The results section lacks a detailed analysis. The authors do not discuss any surprising findings, limitations of the results, or how their results compare to the existing state-of-the-art.
3. The proposed method seems overly complex and less generalized compared to current methods based on large language models. The necessity and practicality of this approach in the era of large models are not convincingly demonstrated.

**Reproducibility:**

3: Could reproduce the results with some difficulty. The settings of parameters are underspecified or subjectively determined; the training/evaluation data are not widely available.

**Reviewer Confidence:**

4: Quite sure. I tried to check the important points carefully. It's unlikely, though conceivable, that I missed something that should affect my ratings.

---

> ### Author Rebuttal · Authors · 2023-08-29
>
> Thank you for your comments and suggestions. Please find below our responses for your questions and concerns.
>
> **Reviewer**: *"The authors do not provide a thorough comparison with existing methods. This leaves the reader unsure of how the proposed method stands against current state-of-the-art methods."*
>
> *"The results section lacks a detailed analysis. The authors do not discuss any surprising findings, limitations of the results, or how their results compare to the existing state-of-the-art."*
>
> **Our Response**:
>
> Thank you for your comments. However, we would like to emphasize that in Section 3 of the paper, we have compared our work with the current state-of-the-art methods for cross-lingual event detection by the time of work for this paper, including XLMR-CRF-CCCAR [1] and  XLMR-OACLED [2]. We hope the reviewer can reconsider this point for us.
>
> Regarding the analysis, we would like to note that we have provided a visualization analysis in Appendix B to demonstrate the transitions of the representations from the target language space to the source language space for our model for cross-lingual event detection. Our analysis shows that the target language representations are gradually shifted toward the source language space based on Langevin Dynamics and the energy function. Importantly, due to the introduction of our semantic preservation mechanism, the final states of the target language representations can also preserve their information about event types that allows the succesful application of the source language to predict event types for the target languages in our method. In addition, in Appendix C and the Limitation Section, we have discussed the limitation of our method regarding additional cost for inference time. Our experiments in Appendix C shows that although our method require additional inference time, this increase is not substantial and it can be accepted given the demonstrated performance benefits.
>
> [1] Nguyen et al., 2021. Crosslingual transfer learning for relation and event extraction via word category and class alignments. In EMNLP 2021.
>
> [2] Guzman-Nateras, et al., 2022. Cross-lingual event detection via optimized adversarial training. In NAACL 2022.
>
> **Reviewer**: *"The proposed method seems overly complex and less generalized compared to current methods based on large language models. The necessity and practicality of this approach in the era of large models are not convincingly demonstrated."*
>
> **Our Response**:
>
> Thank you for your comment. However, as mentioned in our response for reviewer 2, it has actually been shown in some recent work that despite their strong text generation performance, large language models (LLMs), such as ChatGPT, cannot achieve very good performance for event detection and span detection tasks (e.g., named entity recognition) in different languages, e.g., in [3] and [4]. These evaluations show that ChatGPT's performance still significantly falls short of the state-of-the-art supervised learning models for these tasks over different languages with large performance gaps. The main issue might stem from the misalignment between nature of the next word prediction/text generation tasks in the LLMs and the sequence labeling tasks we are deadling in event and span detection. As such, we believe dedicated models for cross-lingual event detection, such as our current work, are still necessary to ensure strong performance for applications. These dedicated models are also more practical given that they don't need gigantic scale to perform well and they can be hosted internally. This will avoid data privacy and security risks as we don't need to send data to external LLMs for processing.
>
> [3] Bo Li, Gexiang Fang, Yang Yang, Quansen Wang, Wei Ye, Wen Zhao, Shikun Zhang. Evaluating ChatGPT's Information Extraction Capabilities: An Assessment of Performance, Explainability, Calibration, and Faithfulness: https://arxiv.org/pdf/2304.11633.pdf
>
> [4] Viet Dac Lai, Nghia Trung Ngo, Amir Pouran Ben Veyseh, Hieu Man, Franck Dernoncourt, Trung Bui, Thien Huu Nguyen. ChatGPT Beyond English: Towards a Comprehensive Evaluation of Large Language Models in Multilingual Learning: https://arxiv.org/pdf/2304.05613.pdf
>
> **Reviewer**: *"1. How does the proposed method compare with existing methods in terms of computational efficiency, especially considering the additional computational cost at inference time for representation transition?"*
>
> **Our Response**:
>
> We would like to note that we have provided an analysis to compare the inference time for our method and previous cross-lingual event detection methods in Appendix C. The results show that our method needs additional time due to the representation transition, but it's not substantial and can be acceptable to obtain strong prediction performance.
>
> **Reviewer**: *"2. Could you elaborate on the limitations of the results? Were there any surprising findings or anomalies in the results that were not discussed in the paper?"*
>
> **Our Response**:
>
> Our extensive experiments demonstrate the benefits of our method for cross-lingual event detection, achieving state-of-the-art performance for different pairs of source and target languages compared to previous work. The ablation study further highlights the necessity and contribution of each component in our representation transition model. The performance improvement is consistent across different languages, and we do not observe any anomalies in the development and performance for our model.
>
> **Reviewer**: *"3. Could you provide more insights into why the proposed method outperforms existing methods? What specific features or aspects of the method contribute to its superior performance?"*
>
> **Our Response**:
>
> We would like to emphasize that we have discussed the limitations of previous work for cross-lingual event detection to motivate the design for our method in this work in L59-87. Essentially, the main approach of the existing methods for cross-lingual event detection is to perform representation matching to transform the representations for the source and target languages into a common space to train language-general ED models. However, to achieve a common space, the source-language representations might need to sacrifice language-specific information, which can involve discriminative features, causing performance degradation for event detection. To address this issue, we introduce a new method for cross-lingual event detection based on representation transition. Instead of transforming the source-language representations, which might eliminate discriminative features for ED, we only seek to transition the target-language representations into the source space whose results can be directly fed into the source-language models for event prediction. In this way, we can preserve the source-language representations with discriminative features learned over training data for ED to maximize prediction ability.
>
> **Reviewer**: *"Given the complexity of the proposed method, how do you see its practical application, especially in real-world scenarios where computational resources might be limited?"*
>
> **Our Response**:
>
> As demonstrated in Appendix C and acknowledged in our Limitation Section, although our method does require more inference time than previous methods for cross-lingual event detection, our analysis demonstrates the additional time is not substantial and it can be accepted for pratical applications to achieve state-of-the-art performance.
>
> **Reviewer**: *"Are there any specific future research directions you suggest based on the findings of this paper?"*
>
> **Our Response**:
>
> As discussed in the Limitation Section of our paper (L372-348), given the additional inference time imposed by our method, future work can explore more efficient approaches for representation transition, aiming to reduce the sampling time during the transition process to improve the efficiency. In addition, future research can explore larger set of languages to obtain better understanding for the generalization of the methods over different languages.

---

### Official Review · Reviewer_dUHy · 2023-08-10

**Soundness:** 2

**Excitement:**

3: Ambivalent: It has merits (e.g., it reports state-of-the-art results, the idea is nice), but there are key weaknesses (e.g., it describes incremental work), and it can significantly benefit from another round of revision. However, I won't object to accepting it if my co-reviewers champion it.

**Paper Topic And Main Contributions:**

The topic of this paper is to develop models for cross-lingual transfor learning for event detection. Previous works focused on representation matching, but they modified the representations for source language examples, which might lose discriminative features for ED. To address this problem, the paper utilize Langevin Dynamics for cross-lingual representation transition and a semantic preservation framework to retain event features, and achieve SOTA performance in CLED benchmaks.

**Questions For The Authors:**

A.Will the framework can also work on other CLTL problems, such as cross-lingual NER?

B.Since Large Language Models (LLM) can achieve great performance in cross-lingual transfer learning, will they also outperform in CLED?

**Reasons To Accept:**

1) The paper leverage Langevin Dynamics for cross-lingual representation transition. This is the first work to explore energy functions and Langevin Dynamics for CLTL in IE and ED.

2) A semantic preservation mechanism is used for the target language examples from Langevin Dynamics to retain event features.

3) Extensive evaluations over three languages demonstrate the paper’s SOTA performance for CLED.

**Reasons To Reject:**

1)The motivation of this paper is not clearly stated. The purpose of this paper is to address issues in previous works where they achieve matching representations but lose discriminative features, potentially leading to impaired performance in Event Detection (ED). However, the presented problem and solution may be questioned. First, prior works typically employ joint optimization for both event detection and representation matching, leading to language-invariant yet event-related representations for ED. Second, sacrificing language-specific information may not be a concern, as the event types in both source and target language datasets are identical. Finally, the paper employs representation transition for Cross-Language Event Detection (CLED). This approach only modifies the representations of the target language, potentially leading to information loss. Unfortunately, the paper does not explicitly compare the pros and cons of representation matching and representation transition.

2)The method proposed in this paper lacks sufficient justification. Firstly, the paper uses an energy function, specifically Langevin Dynamics, for representation transition. However, the paper does not explicitly explain how the energy function impacts representation transition, nor does it clarify why Langevin Dynamics is chosen over other energy functions. Additionally, the techniques employed for semantic preservation are similar to representation matching, making it challenging to comprehend the paper's core motivation and leading to a perception of insufficient innovation.

3)The experimental evaluation in this paper is also inadequate. Firstly, the paper only evaluates the model on the ACE 2005 dataset, focusing on a limited number of language pairs. It disregards other CLED datasets such as ERE[1] and MINION[2], which involve a broader range of language pairs and etymological combinations. Moreover, the ablation study falls short, as it only explores semantic preservation while neglecting the critical aspects of the model: the energy model and Langevin Dynamics, which are essential components deserving of more thorough examination.

[1] Song, Zhiyi, et al. "From light to rich ere: annotation of entities, relations, and events." Proceedings of the the 3rd Workshop on EVENTS: Definition, Detection, Coreference, and Representation. 2015.

[2] Veyseh, Amir Pouran Ben, et al. "MINION: a Large-Scale and Diverse Dataset for Multilingual Event Detection." Proceedings of the 2022 Conference of the North American Chapter of the Association for Computational Linguistics: Human Language Technologies. 2022.

**Reproducibility:**

3: Could reproduce the results with some difficulty. The settings of parameters are underspecified or subjectively determined; the training/evaluation data are not widely available.

**Reviewer Confidence:**

5: Positive that my evaluation is correct. I read the paper very carefully and I am very familiar with related work.

---

> ### Author Rebuttal · Authors · 2023-08-29
>
> Thank you for your comments and suggestions. Please find below our responses for your questions and concerns.
>
> **Reviewer**: *"1)... However, the presented problem and solution may be questioned. First, prior works typically employ joint optimization for both event detection and representation matching, leading to language-invariant yet event-related representations for ED. Second, sacrificing language-specific information may not be a concern, as the event types in both source and target language datasets are identical. Finally, the paper employs representation transition for Cross-Language Event Detection (CLED). This approach only modifies the representations of the target language, potentially leading to information loss. Unfortunately, the paper does not explicitly compare the pros and cons of representation matching and representation transition."*
>
> **Our Response**:
>
> Thank you for your comment. We would like to clarify that while the event types remains consistent between the source and target languages, the linguistic discrepancies give rise to the differences in the representations for the source and target languages as we visualize in Figure 1 of the Appendix. That’s why previous representation matching methods aim to learn a shared representation space for the source and target languages by removing the language-specific features. In our method, we argue that language-specific feature removal might also lead to the **elimination of discriminative features** for event prediction learned from the source-language representations, thus hurting the performance of the systems. As such, our method proposes to avoid language-specific feature removal from the source-language representations by preserving this representation space for the model. Instead, we propose to explicitly transition the representations from the target language space into the source language space to perform event prediction. Our experiment results show that representation transition can produce state-of-the-art performance for cross-lingual event detection over different language pairs, thus testifying to the benefits of our methods and argument.
>
> Regarding the potential information loss due to representation transition, we would like to note that our method introduces a semantic preservation component to explicitly address issue (L195-275). Using representation decomposition with the Variational Auto-Encoder framework, our model seeks to preserve the event type information during the representation transition process to the source language space, thus avoiding information loss for event detection. Our ablation study and performance comparison in Section 3 demonstrates the effectiveness of this component, leading to the state-of-the-art performance for our method.
>
> Finally, as discussed in our Limitation section, compared to the representation matching methods, our representation transition method might require more time during the inference time. In Appendix C, we evaluate the inference time for our methods, showing that the inference time does increase. However, the increase is not substantial and can be acceptable given the demonstrated performance. We will clarify those details in the final version.
>
> **Reviewer**: *"2)The method proposed in this paper lacks sufficient justification. Firstly, the paper uses an energy function, specifically Langevin Dynamics, for representation transition. However, the paper does not explicitly explain how the energy function impacts representation transition, nor does it clarify why Langevin Dynamics is chosen over other energy functions. Additionally, the techniques employed for semantic preservation are similar to representation matching, making it challenging to comprehend the paper's core motivation and leading to a perception of insufficient innovation."*
>
> **Our Response**:
>
> Thank you for your comment. We would like to clarify the role of the energy function in our approach. First, the energy function helps estimate the probability distribution over the source language representations, which is essential to enable our representation transition method. Second, the energy function can be viewed as an implicit generator, enabling us to utilize the Langevin Dynamics algorithm for the purpose of sampling from the distribution to perform representation transition. As such, it is important to note that **Langevin Dynamics is a typical Markov Chain Monte Carlo algorithm (not an energy function)** to generate samples from an initialization. This sample geenration process is leveraged for the representation transition from the target to the source language space in our method. Exploring other data sampling methods (i.e., rather than Langevin Dynamics) can improve our performance further. However, in this short paper, we aim to demonstrate the benefit of our representation transition approach even using a typical sampling method, leaving it for future work to devise better sampling methods for the framework.
>
> Finally, we would like to note some key differences between our semantic presentation component and the representation matching methods. First, our semantic presentation is embedded in a larger framework of representation transition, which is fundamentally different from the representation matching methods. Second, our semantic representation component aims to preserve the event type information in the target language space during the representation transition to ensure correct type prediction with the source language model. The source language representation space learned from training data is preserved in our method. In contrast, the representation matching methods aim to eliminate language-specific features from both source and target spaces to achieve language-universal representations. As such, these methods modify the source language representation space learned from training data, which might cause loss of discriminative features for event detection.
>
> **Reviewer**: *"3)The experimental evaluation in this paper is also inadequate. Firstly, the paper only evaluates the model on the ACE 2005 dataset, focusing on a limited number of language pairs. It disregards other CLED datasets such as ERE[1] and MINION[2], which involve a broader range of language pairs and etymological combinations. Moreover, the ablation study falls short, as it only explores semantic preservation while neglecting the critical aspects of the model: the energy model and Langevin Dynamics, which are essential components deserving of more thorough examination."*
>
> **Our Response**:
>
> Thank you for your comment. We have already performed evaluation for our method over the MINION dataset that provides data for 8 languages English (EN), Spanish (ES), Portuguese (PT), Polish (PL), Turkish (TR), Hindi (HI), Japanese (JP), and Korean (KO). Following the same setting in the original paper of MINION [1], we compare our method RepTran with the state-of-the-art models for MINION in [1], i.e., BERT-CRF and XLMR-CRF, over the cross-lingual performance from English to 7 other languages. Please find our results below:
>
> | Model | EN->ES | EN->PT | EN->PL | EN->TR | EN->HI | EN->JP | EN->KO |
> | :----------- | :------: | :------: | :------: | :------: | :------: | :------: | :------: |
> | BERT-CRF | 62.6 | 71.1 | 59.6 | 46.6 | 58.0 | 34.1 | 55.6 |
> | XLMR-CRF | 62.8 | 72.8 | 60.1 | 47.2 | 58.2 | 35.1 | 56.8 |
> |RepTran (ours) | 65.4 | 75.5 | 62.5 | 49.6 | 59.1 | 38.9 | 59.9 |
>
> As can be seen, our method significantly outperforms the current state-of-the-art performance over MINION, thus further demonstrating its benefits. We will provide this results in the final version.
>
> For the ablation study, we would like to note that the energy model and Langevin Dynamics are critical elements of our representation transition method, and ablating any of them would turn our model back to an simple XLMR model as we cannot obtain any intermediate representations for the transition process and semantic preservation. As the performance of the simple XLMR model (with just the XLMR encoder and a classification head) has been shown to be worse than the current state-of-the-art models, we don't include them in the ablation study. We will clarify this detail in the final version.
>
> **Reviewer**: *"A.Will the framework can also work on other CLTL problems, such as cross-lingual NER? "*
>
> **Our Response**:
>
> Yes, as mentioned in our response to reviewer 1, our representation transition method can also be applied to other cross-lingual transfer learning problems, including the cross-lingual NER task. It has the potentials to preserve discriminative features for the tasks to boost cross-lingual performance. Our current work focuses on cross-lingual event detection and leave the other tasks for future work.
>
> **Reviewer**: *"B.Since Large Language Models (LLM) can achieve great performance in cross-lingual transfer learning, will they also outperform in CLED?"*
>
> **Our Response**:
>
> Thank you for your question. From our understanding, it has actually been shown in some recent work that LLMs, such as ChatGPT, cannot achieve very good performance for event detection and span detection tasks (e.g., named entity recognition) in different languages, e.g., in [2] and [3]. These studies show that ChatGPT's performance is still significantly worse than the state-of-the-art supervised learning models for these tasks over different languages with large performance gaps. The main issue might stem from the misalignment between nature of the next word prediction/text generation tasks in the LLMs and the sequence labeling tasks we are deadling in event and span detection. We believe further exploration of LLMs to overcome such misalignments can lead to promising advancement for this area.
>
> [1] Veyseh, Amir Pouran Ben, et al. "MINION: a Large-Scale and Diverse Dataset for Multilingual Event Detection." Proceedings of the 2022 Conference of the North American Chapter of the Association for Computational Linguistics: Human Language Technologies. 2022.
>
> [2] Bo Li, Gexiang Fang, Yang Yang, Quansen Wang, Wei Ye, Wen Zhao, Shikun Zhang. Evaluating ChatGPT's Information Extraction Capabilities: An Assessment of Performance, Explainability, Calibration, and Faithfulness: https://arxiv.org/pdf/2304.11633.pdf
>
> [3] Viet Dac Lai, Nghia Trung Ngo, Amir Pouran Ben Veyseh, Hieu Man, Franck Dernoncourt, Trung Bui, Thien Huu Nguyen. ChatGPT Beyond English: Towards a Comprehensive Evaluation of Large Language Models in Multilingual Learning: https://arxiv.org/pdf/2304.05613.pdf

---

### Official Review · Reviewer_xAXc · 2023-08-13

**Soundness:** 3

**Excitement:**

3: Ambivalent: It has merits (e.g., it reports state-of-the-art results, the idea is nice), but there are key weaknesses (e.g., it describes incremental work), and it can significantly benefit from another round of revision. However, I won't object to accepting it if my co-reviewers champion it.

**Paper Topic And Main Contributions:**

This paper proposes a representation transition method for zero-shot cross-lingual event detection tasks. The goal is to align representation spaces among languages and also preserve the discriminative information. Specifically, the Langevin Dynamics is leveraged to perform representations for the target language. Experimental results demonstrate the effectiveness of three languages. Also the authors claim that this work is the first to explore energy functions in the information extraction and event detection.


**Questions For The Authors:**

1. Whether the proposed method can also work on other cross-lingual tasks such as tasks in the XTREME benchmark?

2. In Line 356, It is true that “data project requires high-quality machine translation” and have you compared your methods with any translation-test baselines?


**Reasons To Accept:**

1. small and focused contribution of improving zero-shot cross-lingual event detection.

2. Clear writing and good representation.

3. Achieve state-of-the-art performance


**Reasons To Reject:**

It seems that the proposed solution can not well support the motivation. The general idea is to disentangle the language-specific and task-specific features from the representations. It is not clearly stated or explained why the proposed solution can achieve this while other alignment based methods fail. It is suggested to justify some insights upon this.


**Reproducibility:**

4: Could mostly reproduce the results, but there may be some variation because of sample variance or minor variations in their interpretation of the protocol or method.

**Reviewer Confidence:**

4: Quite sure. I tried to check the important points carefully. It's unlikely, though conceivable, that I missed something that should affect my ratings.

---

> ### Author Rebuttal · Authors · 2023-08-29
>
> Thank you for your comments and suggestions. Please find below our responses for your questions and concerns.
>
> **Reviewer**: *"It seems that the proposed solution can not well support the motivation. The general idea is to disentangle the language-specific and task-specific features from the representations. It is not clearly stated or explained why the proposed solution can achieve this while other alignment based methods fail. It is suggested to justify some insights upon this.
> "*
>
> **Our Response**:
>
> Thanks for your comment. However, we would emphasize that our main motivation is to transition the representations from the target-language space to the source-language space using Langevin Dynamics for event detection. The decomposition of representations into language-invariant and language-specific features is just a component of our representation transition proposal to facilitate semantic presentation during the transition process. This is juts in contrast to previous cross-lingual models for event detection, e.g. adversarial training, that focus on learning a universal representation space for the source and target languages by eliminating language-specific features to align the source- and target-language representations. As discussed in our paper, the representation alignment in previous method might also remove discriminative features from the source-language representations for event prediction, thus hurting the performance. Our representation transition method does not suffer from this issue as we preserve the representation space for the source languages (i.e., we only shift the target-language representations), thus maintaining discriminative features for event detection.
>
> **Reviewer**: *"Whether the proposed method can also work on other cross-lingual tasks such as tasks in the XTREME benchmark?"*
>
> **Our Response**:
>
> Yes, our representation transition method can also be applied to other cross-lingual tasks, including those in the xTREME benchmark. It has the potentials to preserve discriminative features for the tasks to boost cross-lingual performance. Our current work focuses on cross-lingual event detection and we will explore other tasks in future work.
>
> **Reviewer**: *"In Line 356, It is true that “data project requires high-quality machine translation” and have you compared your methods with any translation-test baselines?"*
>
> **Our Response**:
>
> Our intuition is that the translation-based methods are not comparable to our representation transition model as the transition-based methods would require translation and text alignment systems while our method does not need those resources. A comparison will be not fair in this case. In addition, for translation-based methods, it is expected that the translation and alignment systems should have high quality to avoid noise for the data, which will introduce another challenge for this method and limit their applications to many other languages.

---

### Meta-Review · Area_Chair_2Dxc · 2023-10-05

**Recommendation:** 3

**Metareview:**

This paper proposes method for cross-lingual event detection. The method has promise for solving the cross-lingual event detection task, performing zero-shot event detection; experiments suggest improvement over the state-of-the-art models.
It needs a refinement with respect to the following: generalizability of the method for the other datasets; comparisons with more potential techniques; and a more clearly stated motivation.

---

### Decision · Program_Chairs · 2023-10-07

**Decision:**

Accept-Findings

**Comment:**

This paper proposes method for cross-lingual event detection. The method has promise for solving the cross-lingual event detection task, performing zero-shot event detection; experiments suggest improvement over the state-of-the-art models.
It needs a refinement with respect to the following: generalizability of the method for the other datasets; comparisons with more potential techniques; and a more clearly stated motivation.